# Sexual Dimorphism in the Skeletal Morphology of Asian Elephants (*Elephas maximus*): A Preliminary Morphometric Study of Skull, Scapula, and Pelvis

**DOI:** 10.3390/biology14080933

**Published:** 2025-07-24

**Authors:** Piyamat Kongtueng, Promporn Piboon, Sarisa Klinhom, Intorn Aunsan, Nontanan Tongser, Taweepoke Angkawanish, Korakot Nganvongpanit, Burin Boonsri

**Affiliations:** 1Faculty of Veterinary Medicine, Chiang Mai University, Chaing Mai 50100, Thailand; piyamat.k@cmu.ac.th (P.K.); promporn.piboon@cmu.ac.th (P.P.); yui.sarisarisa@gmail.com (S.K.); intorn.u@cmu.ac.th (I.A.); nontanan.t@cmu.ac.th (N.T.); korakot.n@cmu.ac.th (K.N.); 2Thai Elephant Conservation Center, Lampang 52190, Thailand; taweepoke@gmail.com

**Keywords:** cranium, ossa coxae, scapulae, sex differences

## Abstract

This early study looked at bones from Asian elephants to see how male and female skeletons differ. The researchers measured the skull, shoulder blade (scapula), and hip bone (pelvis) from eleven elephant skeletons using basic tools like tape measures and calipers. They found no major differences between males and females in the skull, but they did find clear differences in the hip and shoulder bones. The hip bone showed the greatest differences, especially in the overall length of the pelvic girdle, the length of the pubic symphysis where the two hip bones join, and the minimum width of the pubic shaft. The shoulder blade also showed differences in width, diagonal size, and the length of the ridge on the bone. When the team used the hip bone measurements to determine whether the skeleton belonged to a male or female, they were right 91% of the time—getting all the female cases correct and two-thirds of the male ones. This shows that the hip bone has the best data for telling apart male and female elephant skeletons. These results could help with wildlife studies, museum work, or identifying remains when only bones are available.

## 1. Introduction

As the largest land mammals, elephants have unique skeletal structures specifically designed to support their enormous size and weight. In-depth knowledge of these anatomical features is crucial for precise species identification, health evaluations, and the formulation of effective conservation strategies [1]. A comprehensive study of elephant anatomy is essential for a variety of scientific and practical applications, such as wildlife conservation, veterinary medicine, evolutionary research, and forensic investigations [2,3]. Research into elephant anatomy has a long history, with anatomists establishing a foundation for our current understanding. In particular, the skull, scapula, and pelvis provide valuable insights due to their unique morphological characteristics and function [4,5,6].

Sexual dimorphism in skeletal anatomy refers to sex-related differences in bone size, shape, and proportion, which commonly reflect distinct reproductive roles and life-history strategies [7,8,9,10,11,12,13,14]. In elephants, sexual dimorphism is well-documented in features such as body size and tusk development [15]. However, there remains a need for more detailed investigations into skeletal differences between sexes, particularly through morphometric approaches. In humans, males have larger absolute pelvic dimensions because of their greater overall body size, yet females possess relatively wider pelvic inlets and outlets, a difference produced by variations in the size, shape, and orientation of the ilium and ischiopubic rami that accommodate childbirth [16]. In dogs, distinct sexual differences have also been observed in specific skeletal features, including the coronoid process of the mandible, the caudal ventral iliac spine, and the angle of the ischiatic arch [10]. Similar dimorphic patterns that have been reported in felines showed that a combination of pelvic and mandibular measurements, together with three easily recognizable bony hallmarks, could predict sex with up to 97% accuracy [11]. In dugongs, skull morphometrics enabled sex identification with 96.9% accuracy, and the presence of large permanent tusks provided 100% accuracy. Furthermore, skull measurements distinguished dugong populations between the Andaman Sea and the Gulf of Thailand with perfect accuracy [12]. Another study also highlighted that, in mature dugongs, male pelvic bones are larger than those of females, provide high predictive accuracy for sex determination, and show the strongest correlation with body length, followed by body weight and age [14]. In the African elephant, the pelvic floor is flat in females, while in males, it has a dorsal inclination at the level of the obturator foramen, indicating differences in pelvic structure [5].

Traditional morphometric methods are useful for quantifying these differences, as they allow for the precise measurement of various skeletal elements, thereby clarifying the subtle variations that may not be apparent through only qualitative observations.

This study was to determine a comprehensive anatomical analysis of the elephant’s skeletal structures, especially focusing on the skull, scapula, and pelvis. The research aimed to investigate the potential for sexual dimorphism within these anatomical features by applying traditional morphometric measurements to the skeletal remains in museums. Despite these constraints, examining these specimens provides valuable insights into the skeletal dimorphism and contributes to our understanding of anatomical variations between male and female elephants.

## 2. Materials and Methods

### 2.1. Sample

The skeletal samples of Asian elephants (*Elephas maximus*) used in this study were obtained from the Veterinary Anatomy and Pathology Museum at the Faculty of Veterinary Medicine, Chiang Mai University, Thailand, and the museum at Thai Elephant Conservation Center, Lampang, Thailand. Samples identified as juvenile elephants, based on their identification certificates, were excluded from this study. For bones without identification certificates, age classification was determined through morphological assessment. Specifically, scapular samples were considered adult if they exhibited no visible signs of ossification, while skull samples were classified as adult if the sutures between the parietal bones were fully fused and the ratio of skull length to height was approximately 0.75:1 [17]. A total of 11 samples, comprising 9 dried skulls, 11 dried pelves, and 18 dried scapulae (7 paired, 2 right scapulae, and 2 left scapulae), were used for morphometric measurements.

### 2.2. Anatomical Measurement

Measurements were obtained using a measuring tape when a direct straight line between the two measurement points was accessible. For points where the bone’s curvature prevented a direct measurement, large calipers were employed to ensure accuracy. Alternatively, two flat surfaces were positioned at the designated measurement points, provided they extended sufficiently to allow the tape to be stretched in a straight line between them. A list of all measurements (Table 1, Figure 1) was adapted from a previous study involving elephant subjects [5,6,18,19].

### 2.3. Study Design and Statistical Analysis

This research comprised two studies, as outlined below. Continuous variables are summarized and presented as mean values with standard deviations (Mean ± SD) to describe the relevant morphometric data. *p* < 0.05 was considered statistically significant. Specific statistical methodologies were employed for each study. The descriptive and correlation analyses were performed using R software (version 3.6) via Posit team (2024). (RStudio: Integrated Development Environment for R. Posit Software, PBC, Boston, MA, USA, http://www.posit.co/, accessed on 1 June 2025). Data visualization, logistic regression analysis, and the generation of Area Under the Curve (AUC) values were conducted using DATAtab (2025) (DATAtab e.U., Graz, Austria; available at https://datatab.net, accessed on 1 June 2025). The missing values are indicated as “N/A” in the Appendix A and were excluded from statistical analyses to maintain accuracy and consistency in the results.

#### 2.3.1. Study 1: Descriptive Analysis and Correlation of Morphometric Data for Skull, Scapula, and Pelvis

Morphometric measurements of the skull, scapula, and pelvis were analyzed and reported separately for three groups: males, females, and individuals of unknown sex. In comparison between groups, the Mann–Whitney U test was performed. To explore relationships between morphometric parameters, Pearson’s correlation analysis was performed. The resulting correlations were visualized as a heatmap, providing an intuitive overview of the strength and direction of the associations among variables.

#### 2.3.2. Study 2: Sex Prediction Using Morphometric Data

For each morphometric variable, we compared male and female values using the Mann–Whitney U test. All measurements from both sexes were first ranked together; the summed ranks for the two groups were then evaluated to determine the degree of overlap, with *p* < 0.05 taken as significant. Only those variables that met this significance threshold were subsequently combined into ratio indices and entered into the logistic regression models for sex prediction [10]. Logistic regression analysis was employed to develop predictive models for determining sex and assessing skull shape. The performance of these models was evaluated using metrics such as the accuracy rate and the Area Under the Curve (AUC), providing insights into the reliability and precision of the models for sex classification.

## 3. Results

A total of 11 elephant dry skeletal specimens (CM-01 to CM-11) were received from the Veterinary Anatomy and Pathology Museum at the Faculty of Veterinary Medicine, Chiang Mai University, Thailand. Of these, five specimens had identifiable information, with a mean age at death of 56 years. A total of nine skulls were examined, comprising two males, five females, and two of unknown sex (Appendix A; Figure 2). The scapulae were analyzed on both right and left sides. The right scapula group consisted of four males and five females (Appendix A; Figure 3), while the left scapula group included three males and six females (Appendix A; Figure 4). For the pelvis, all 11 samples were analyzed, including three males and eight females (Appendix A; Figure 5). The pelvis was the only bone with complete measurements and available for sex discrimination analysis. No significant differences were observed in skull measurements between sexes. Conversely, measurements of the pelvic girdle revealed significant differences. Specifically, the length of the pelvic girdle (LPG), the length of the symphysis (LPS), and the minimum perimeter of the pubis shaft (PPS) exhibited statistical significance, with *p*-values of 0.024, 0.012, and 0.048, respectively. For the right scapula, significant differences were found in the measurements of the diagonal breadth (DBL-R), the mediolateral width (MLV-R), and the length of the scapular spine (LSS-R), with *p*-values of 0.016, 0.016, and 0.032, respectively. In the same way, for the left scapula, the diagonal breadth (DBL-L), mediolateral width (MLV-L), width at the scapular head (WSH-L), and the length of the scapular spine (LSS-L) also demonstrated statistical significance, with *p*-values of 0.024, 0.048, 0.048, and 0.024, respectively.

The results of the correlation analysis are presented in the form of a heatmap (Figure 6). The Pearson correlation heatmaps reveal several statistically significant clusters (r > 0.70, *p* < 0.05). In the skull (Figure 6A), proximal and distal premaxillary widths, tusk-socket circumferences, condylo-basal length, and condylo-zygomatic length form a tightly linked group, indicating coordinated growth of the facial and basicranial regions. In the pelvis (Figure 6B), horizontal widths (HWPG, HWPA, HWA) correlate strongly with weight-bearing landmarks (WEI) and with the length-based cluster LPG–LPS–PPS, suggesting integrated load-distribution mechanisms. For the right scapula (Figure 6C), the dorsal and cranial border lengths cluster with maximum dorsal length, while the caudal border length aligns with scapular-spine and suprahematus-process lengths—patterns consistent with muscle-attachment zones. The left scapula (Figure 6D) shows an analogous cranial/caudal border cluster linked to glenoid-cavity length and a second cluster centered on the supraglenoid length that correlates with neck and head widths. These clusters identify functionally and developmentally integrated regions within each bone.

As part of the sex prediction analysis, consistent with most mammalian studies, there is no single skeletal measurement in the dataset that reliably separates the sexes. Accordingly, we applied logistic regression to the complete pelvic dataset (the only bone with full male and female records) to assess how well different variables, individually and in combination, addressed the usual trade-off between specificity (correct identification of one sex) and sensitivity (correct identification of the other). Among the single-ratio models (Table 2 and Table 3; Figure 7A–C), LPG/LPS, LPG/PPS, and LPS/PPS each achieved perfect specificity for females (100%) but low sensitivity for males (33.3%), resulting in an overall accuracy of 81.8%. Pairing two ratios (LPG/LPS with LPG/PPS or LPG/LPS with LPS/PPS) improved male sensitivity to 66.7% while female specificity remained high at 87.5%, and total accuracy did not change (Table 4 and Table 5; Figure 7D,E). The best performance came from using all three ratios together (Table 6 and Table 7; Figure 7F); overall accuracy rose to 90.9%, female classification stayed perfect, and male sensitivity increased modestly to 66.7%. These findings illustrate a common pattern in morphometric sex diagnosis, where single measurements strongly identify one sex, but multi-parameter models provide a more balanced and therefore more practical classification tool. 

## 4. Discussion

This study provides anatomical insight into the skeletal morphology of Asian elephants (*Elephas maximus*), emphasizing the potential for sexual dimorphism through traditional morphometric analysis. Among the bones studied, the pelvis emerged as the most reliable structure for distinguishing sex, particularly when employing combined parameter models in logistic regression. While the skull showed consistent size differences between sexes, these did not reach statistical significance, underscoring the subtlety of cranial dimorphism. In contrast, both the pelvis and scapula demonstrated significant differences in specific parameters, reinforcing their functional and structural roles in supporting body mass and locomotion. Correlation analysis further revealed distinct anatomical clusters within each bone, suggesting coordinated morphometric relationships and highlighting parameters that may serve as integrative structural indicators. Collectively, these findings establish a critical reference point for future comparative and forensic studies and support the application of morphometric methods in the sex identification of elephant skeletal remains.

The skull, which is a highly prominent and structurally complex bone, exhibited variations in landmark dimensions in both males and females. In a previous study in *Loxodonta africana*, the sex was determined based on the skull morphology [20]. The crania of male elephants were larger than those of females. For instance, the tusk alveoli and occipital condyles in males were larger compared to females. In the same study, the Asian elephant was also analyzed, revealing differences in muscle markings, attachment sites, and the parietal–occipital crest between sexes.

In our study, morphometric data were collected and analyzed with the aim of providing explicit information on sexual dimorphism. However, the differences in mean values across several parameters were relatively subtle, reflecting the overall robustness of elephant skulls regardless of sex. The circumference of the internal tusk sockets, associated with the distal premaxillary width, appeared to be larger in males than in females. Similarly, overall measurements of the cranium, including the condylo-basal length, condyle-zygomatic length, and zygomatic width, were also observed to be larger in males compared to females. We presumed that the evolutionary factors may mute cranial dimorphism in Asian elephants. First, both sexes need equally robust skulls to support tusk alveoli, extensive sinuses, and strong trunk–muscle attachments, creating shared biomechanical constraints. Second, sexual selection appears to target tusk size rather than overall skull shape, so cranial proportions remain similar while tusks diverge. Lastly, strong allometric scaling with body mass means that, once size is accounted for, little shape variation is left to express sex differences. Although the observed variation between sexes was not statistically significant, the data provide baseline anatomical information that could be valuable for future comparative studies. In sex determination, the use of geometric morphometric techniques may yield different results compared to traditional morphometric methods.

The correlation analysis for the skull, in our study, highlights a significant cluster involving proximal premaxillary width, distal premaxillary width, circumference inside tusk sockets, condyle-basal length, and condyle-zygomatic length, with these measurements showing strong intercorrelations. This suggests that they may be part of an integrated system or share underlying anatomical or functional characteristics. Premaxillary length is also highly correlated with these variables, indicating that rostro-caudal elongation of the premaxilla scales proportionally with widening of the tusk region and expansion of the basicranium. In other words, premaxillary length acts as a longitudinal axis that developmentally coordinates the size of the surrounding facial and basal skull elements. The cephalic index (CI) is a key parameter used to differentiate between human populations and various skull types in animal species, such as brachycephalic and dolichocephalic in dogs [21,22]. It is particularly useful in explaining anatomical variations that may influence the health status of these animals [23,24,25]. In comparison to our study, the condyle-basal length and condyle-zygomatic length could potentially be used to calculate the CI, thereby aiding in the classification of elephant skull types. The distinct correlation between cranial width, post-orbital width, and nares width implies that they may form a separate subsystem within the skull structure, with post-orbital width and nares width showing a potential sub-cluster relationship. The condyle-zygomatic length emerges as a highly interconnected parameter, correlating with multiple measurements (NW, pPMW, dPMW, CTSR, CTSL, and CZL), positioning it as a central or mediating factor within the structural network. The circumference inside the tusk sockets and the condylo-basal length appear pivotal, linking the premaxillary widths to condylo-zygomatic and zygomatic widths. Because all of these variables are linear dimensions, their tight correlations are more plausibly driven by overall skull size scaling than by a specific functional interaction. Nonetheless, this size-related coherence helps maintain structural integrity across the expanding facial and basicranial regions.

Beyond identifying skull types using the cephalic index, this approach could highlight distinctive cranial features among individual elephants. These features could also be developed as a tool for assessing interpopulation and intrapopulation morphological characteristics of the elephant head and face, similar to the standardized terminology used in human phenotypic studies [26]. Although we did not calculate the cephalic index (CI) in this preliminary study, the CI is a simple ratio that, despite substantial overlap between sexes, can illustrate subtle skull-shape differences within and between elephant populations, and therefore merits investigation in future, larger samples.

The scapulae, mean lengths of key landmarks, showed slight but consistent differences between the sexes, with males generally exhibiting larger scapulae. This aligns with the notion that male elephants tend to have more robust skeletal features to support their larger body mass.

Despite this, the variation in scapular dimensions suggests that the degree of sexual dimorphism in this bone may be less pronounced compared to other skeletal regions, such as the pelvis. These findings are particularly valuable in forensic science, offering a reliable method for post-mortem identification in cases of natural disasters or criminal investigations [27]. In our study, the lengths from the Angulus cranialis to Angulus caudalis and from the Angulus cranialis to the Supraglenoid tubercle, along with the anterio-posterior width at the scapular glenoid, were consistently larger in males compared to females. These findings suggest that male elephants possess more robust scapulae, which may reflect the demands of supporting their greater body mass and muscle attachment requirements. The observed differences underscore the scapula’s functional adaptations in locomotion and its potential role in distinguishing between sexes in morphometric studies. These significant distinctions enhance the value of scapular measurements as a complementary tool for sex determination in elephants.

The scapular analysis reveals significant findings for both the right and left scapulae. Because 11 of the 18 scapulae formed true right–left pairs while the remaining four specimens were isolated elements, we analyzed right and left scapulae separately. Independent treatment is further justified by a substantial body of osteological research showing that limb bones display measurable directional asymmetry [28]. On the right scapula, very high correlations between the dorsal border length, cranial border length, and maximum length of the dorsal part indicate a tightly interconnected group, suggesting these regions may share structural or functional dependencies. Similarly, the caudal border length has a very high correlation with the length of the scapula spine, and its high correlation with the length of the suprahematus process indicates a trend linked to a specific anatomical feature.

The left scapula displays a pattern where the cranial and caudal border length and the length of the glenoid cavity are highly correlated, suggesting shared anatomical characteristics or functional roles. The length of the supraglenoid acts as a central parameter on the left scapula, correlating highly with various parameters (DBL-L, CrBL-L, CaBL-L, and LGC-L), indicating its role as a unifying measurement. This pattern suggests that LS-L could be pivotal for structural stability or functional integration. The consistent high and very high correlations observed among specific parameters support the notion of a coordinated architectural or biomechanical function. The relationships involving the parameters including CrBL-L, CaBL-L, LS-L, LGC-L, and WSN-L may reflect areas critical for muscle attachment or stress distribution necessary for movement and stability.

Among the three bones studied, some parameters of the pelvis showed significant differences between males and females. Measurements of key landmarks revealed consistent patterns of sexual dimorphism, with certain parameters distinctly larger in males or females. The clear differentiation in pelvic dimensions provides critical support for using this bone as a reliable predictor of sex in elephants, as evidenced by the logistic regression analysis conducted in this study.

The correlation analysis of the pelvis reveals significant interdependencies among various anatomical parameters, highlighting the structural and functional cohesion of the pelvic girdle in elephants. The strong positive correlation between the Horizontal Width of the Pelvic Girdle (HWPG) and the Minimum Perimeter of the Pubis Shaft (PPS) suggests a biomechanical link, potentially reflecting shared roles in weight distribution and locomotor support. Given that the pubis contributes to the structural integrity of the pelvic floor, this correlation may indicate its role in stabilizing the pelvis during movement. Additionally, the high correlation between the length of the pelvic girdle (LPG) and both the Width of the Ilium Wing (WIW) and the length of the symphysis (LPS) suggests a cohesive anatomical unit where these dimensions are interrelated. This cluster (LPG, WIW, and LPS) likely represents a morphometric relationship influenced by similar developmental and functional factors. The ilium, a key load-bearing structure in elephants, plays a crucial role in transferring weight from the spine to the hind limbs. The observed correlation patterns may indicate that variations in these dimensions contribute to optimizing load-bearing capacity, balance, and locomotor efficiency.

The pelvis serves two primary functions in terrestrial mammals: providing support for locomotor muscles and, in females, functioning as the birth canal [29]. The shape of the pelvis is closely linked to body mass and locomotion, with the iliac crest and acetabulum exhibiting the most notable differences among species. In elephants (*Elephas maximus*), the pelvis is more upright than in lighter animals, allowing it to support substantial body weight without dislocating the sacroiliac joint or placing excessive torsional stress on the vertebral column [30].

Elephants possess a distinctive skeletal structure that is essential for their locomotion, with key bones involved including the scapula, humerus, femur, tibia–fibula, carpals, and tarsals. The pelvis, in particular, is a crucial element of their skeletal system, playing a significant role in supporting movement and maintaining stability. It not only facilitates locomotion but also enhances energy efficiency, contributing to the characteristic gait patterns of elephants [31]. Future research could focus on expanding the sample size and incorporating a broader range of age groups and sexes to further refine the understanding of pelvic morphometry in elephants. Advanced imaging techniques such as CT scanning or 3D modeling could provide more detailed structural insights into pelvic adaptations and biomechanical functions.

Sex determination using pelvic morphology has been extensively studied across human populations, demonstrating the reliability of pelvic measurements in distinguishing between sexes [32,33,34,35,36]. A comprehensive review study of pelvic sex determination confirms that the pelvis is one of the most sexually dimorphic skeletal structures, achieving accuracy rates of up to 95%. However, no single method, whether morphological, morphometric, or radiological, is sufficient on its own. A combination of techniques is recommended to ensure consistent and reliable results [35]. Additionally, skeletal traits vary among populations, underscoring the need for population-specific standards when applying morphometric analyses. These insights are particularly relevant to sex dimorphism research in Asian elephant pelvises, where traditional morphometric methods must be carefully evaluated and refined to maximize accuracy. Our findings suggest that, while individual parameters may have varying degrees of predictive accuracy, combined metrics can enhance the overall reliability of sex prediction in morphometric studies. The results emphasize the pelvis’s utility in sex determination, particularly when employing multi-parameter models. The observed disparity in male prediction rates indicates potential factors affecting sensitivity, warranting further investigation into the anatomical and sample size-related influences on male classification.

One significant limitation of this study is the relatively small sample size used for logistic regression analysis. The limited number of samples can affect the robustness and generalizability of the predictive model, as it may not capture the full variability present within the broader population. This constraint can lead to potential biases in the results, particularly impacting the sensitivity for male identification, which was lower in several parameter combinations. The reduced sample size may contribute to overfitting or underpowered statistical conclusions, limiting the confidence in applying these findings universally. Future studies should assemble larger datasets to improve the reliability of sex-prediction models, capture the natural variability in elephant skeletal measurements, and enable multivariate techniques such as factor analysis to disentangle size effects from shape-related clusters that may underlie sexual differences. Additionally, integrating advanced imaging technologies, such as CT scans and 3D surface scanning, could significantly improve the precision and reproducibility of morphometric analyses [37,38,39,40]. These technologies allow for more detailed landmark identification and volumetric assessments, reducing measurement errors and enhancing the ability to detect subtle morphological differences. The use of digital models also facilitates data sharing and comparative studies across different populations, ultimately contributing to the refinement of sex determination methods in elephant osteology.

## 5. Conclusions

This study provides a foundational anatomical profile of the Asian elephant (*Elephas maximus*) skeleton, highlighting the pelvis as the most reliable bone for sex differentiation using traditional morphometric analysis. While the skull and scapula exhibited some sex-related differences, only pelvic measurements demonstrated statistically significant dimorphism and predictive value in logistic regression models—especially when combined parameters were used. Correlation analyses across all bones revealed coordinated morphometric structures, offering insights into functional anatomy and biomechanical adaptation. Despite limitations due to sample size, the findings underscore the value of quantitative morphometry for sex determination and comparative anatomical research. Future studies incorporating larger datasets and advanced imaging technologies are recommended to enhance the precision and applicability of these methods in elephant osteology and conservation science.

## Figures and Tables

**Figure 1 biology-14-00933-f001:**
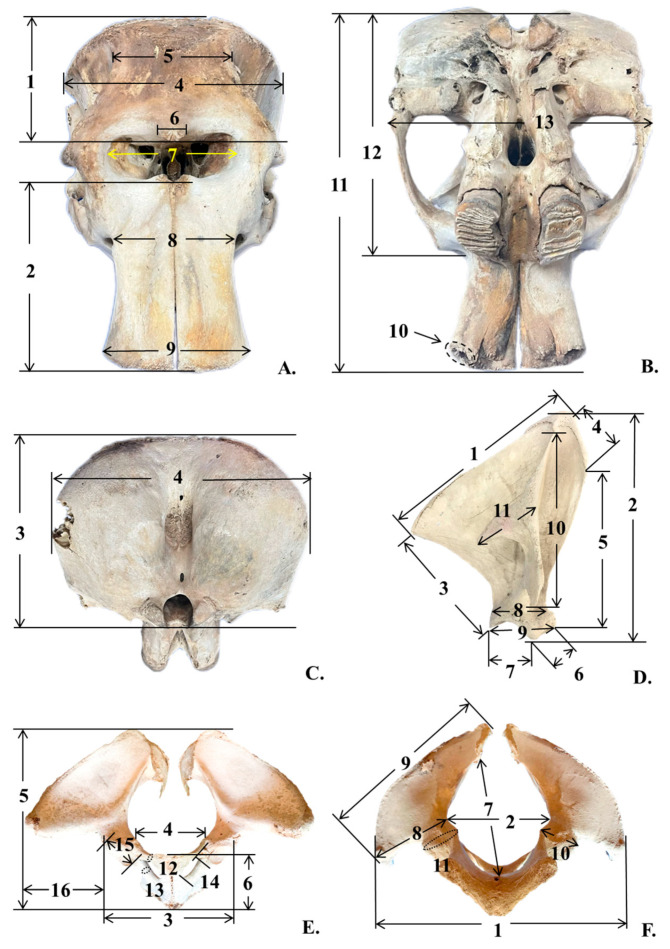
This figure illustrates the traditional morphometric measurements used for analyzing the skull of an Asian elephant: (**A**) Anterior view of the skull. (**B**). Base of the skull. (**C**). Posterior view of the skull. (**D**). Lateral view of the right scapular. (**E**) Anterior view of the pelvis. (**F**) Posterior view of the pelvis.

**Figure 2 biology-14-00933-f002:**
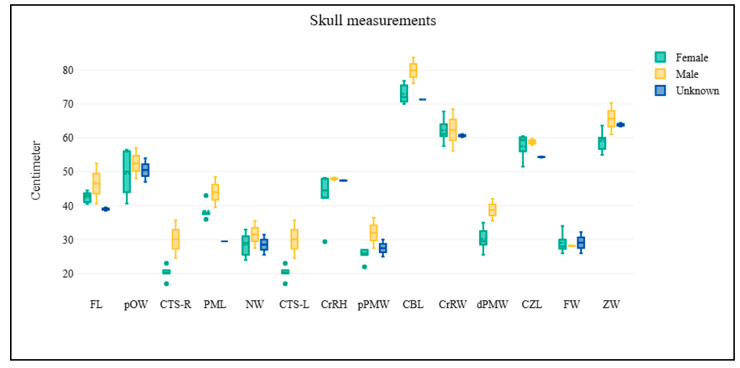
The figure presents a comparative analysis of skull measurements among male, female, and unknown-sex Asian elephants. The data is visualized using box plots, where yellow represents males, green represents females, and blue represents individuals of unknown sex. Each parameter on the *x*-axis corresponds to specific morphometric measurements of the skull, while the *y*-axis indicates the measured values in centimeters. The results indicate substantial overlap in all skull measurement parameters between sexes, with no statistically significant differences observed.

**Figure 3 biology-14-00933-f003:**
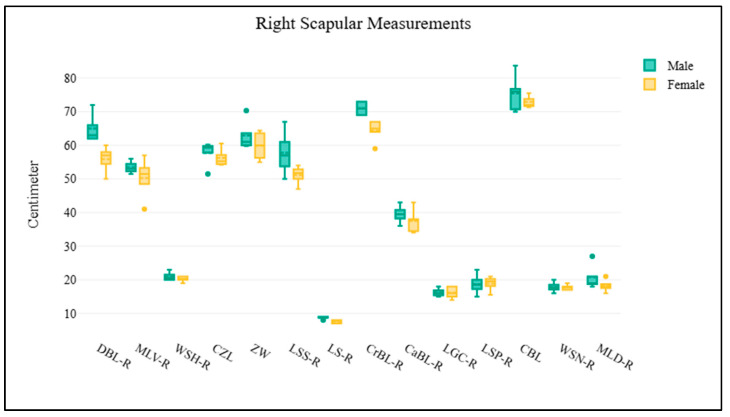
The morphometric data of the right scapula is visualized using box plots. Each parameter on the *x*-axis corresponds to specific morphometric measurements of the right scapula, while the *y*-axis indicates the measured values in centimeters. The results demonstrate that males exhibit larger mean values across most parameters; however, there is no difference between sex.

**Figure 4 biology-14-00933-f004:**
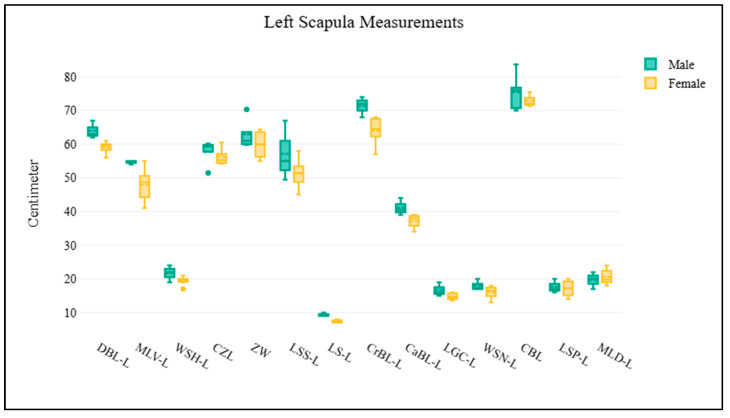
The morphometric data of the left scapula is visualized using box plots. Each parameter on the *x*-axis corresponds to specific morphometric measurements of the left scapula, while the *y*-axis indicates the measured values in centimeters. The results demonstrate that males exhibit larger mean values across most parameters; however, there is no difference between sex.

**Figure 5 biology-14-00933-f005:**
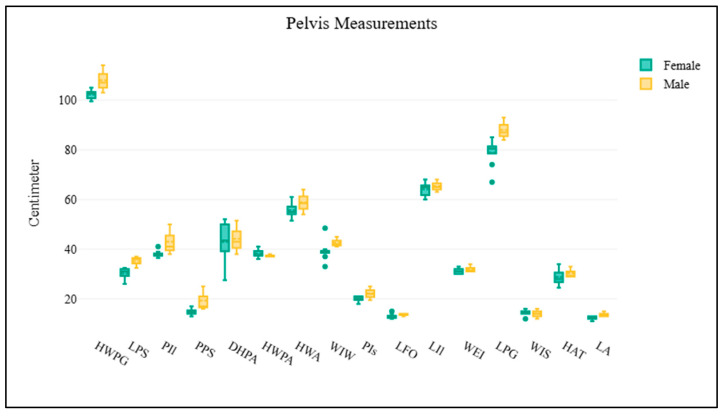
The figure illustrates a comparison of pelvic measurements between male and female Asian elephants. Box plots are used to represent the data, with yellow denoting males and green representing females. The *x*-axis displays various morphometric parameters of the pelvis, while the *y*-axis shows their corresponding measurements in centimeters. No statistically significant differences are identified between the groups.

**Figure 6 biology-14-00933-f006:**
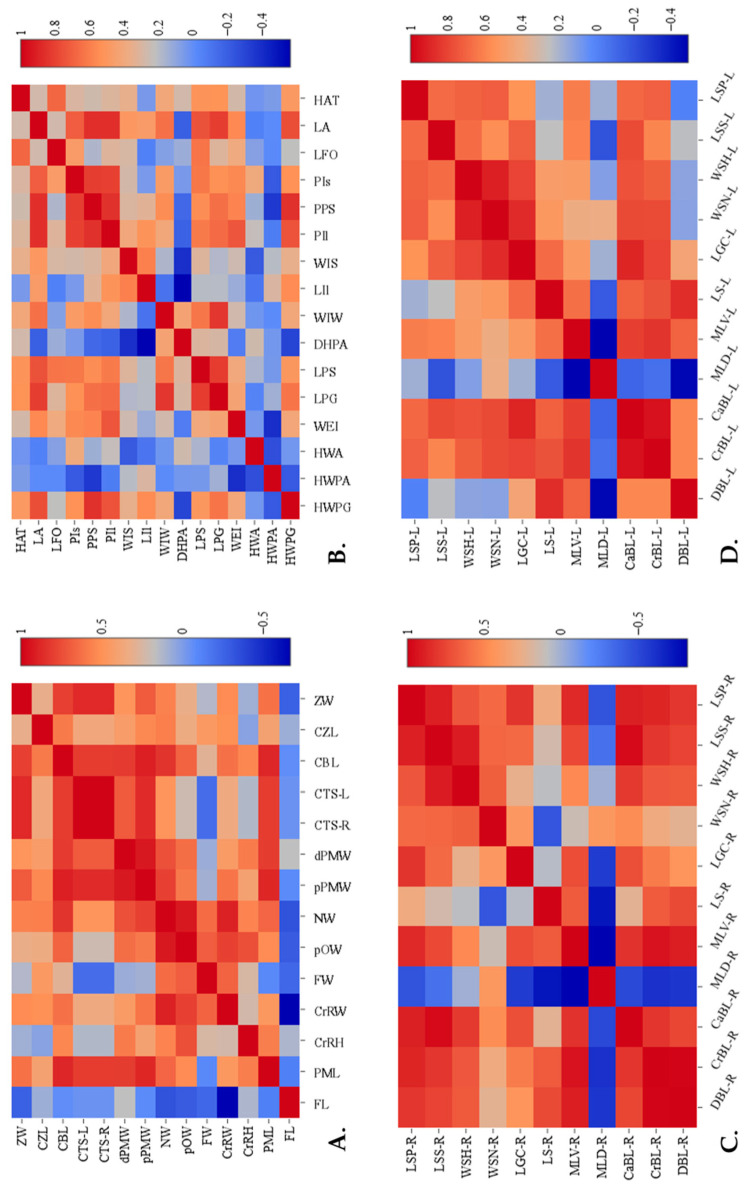
Heatmaps representing the Pearson’s correlation coefficients among morphometric parameters of elephant bones: (**A**) Skull measurements. (**B**) Pelvic measurements. (**C**) Right scapular measurements. (**D**) Left scapular measurements. The scale bar at the right of each heatmap represents correlation coefficient values ranging from −1 to +1, where deeper red tones signify stronger positive correlations, and darker blue tones indicate stronger negative correlations.

**Figure 7 biology-14-00933-f007:**
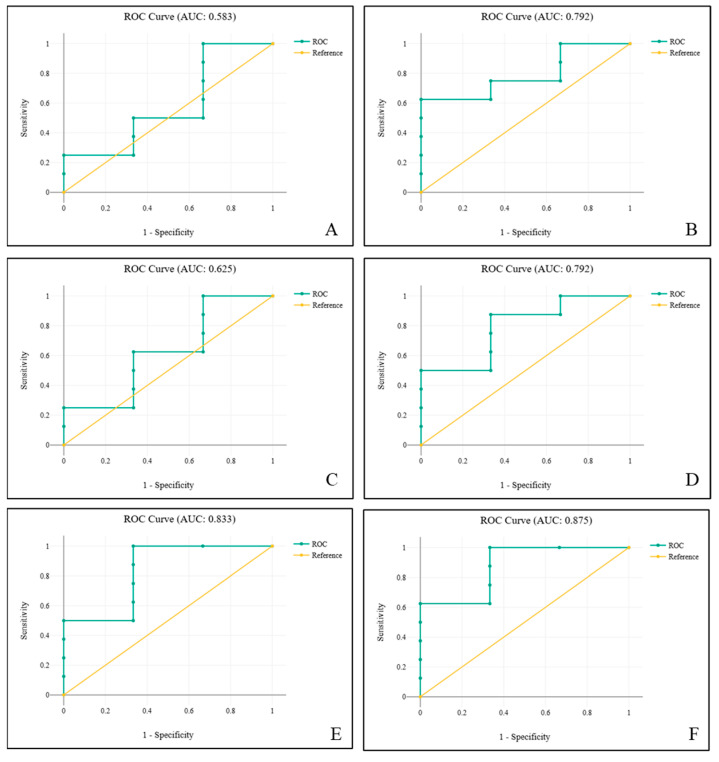
The Receiver Operating Characteristic (ROC) curves evaluating the performance of logistic regression models for sex prediction in elephants using pelvic bone morphometric combination ratios: (**A**) LPG/LPS. (**B**) LPG/PPS. (**C**) LPS/PPS. (**D**) LPG/LPS + LPG/PPS. (**E**) LPG/PPS + LPS/PPS. (**F**) LPG/LPS + LPS/PPS + LPG/PPS. The Area Under the Curve (AUC) values indicate the predictive performance of each model, reflecting discrimination between sexes. The highest predictive accuracy is observed in (**F**) (AUC = 0.875), followed by (**E**) (AUC = 0.833) and (**B**,**D**) (both AUC = 0.792), suggesting improved performance when multiple morphometric ratios are combined. In contrast, the individual predictor LPG/LPS (**A**) shows the lowest discrimination ability (AUC = 0.583).

**Table 1 biology-14-00933-t001:** Morphometric measurements taken from skull, pelvis, and scapula.

Number	Parameter	Abbreviation	Key
**1** **2** **3** **4** **5** **6** **7** **8** **9** **10-R** **10-L** **11** **12** **13**	*Skull* Frontal length Premaxillary length Cranial height Cranial width Frontal width Post-orbital width Nare width Proximal premaxillary width Distal premaxillary width Circumference inside tusk socket (Right) Circumference inside tusk socket (Left) Condylo-basal length Condylo-zygomatic length Zygomatic width	FL PML CrRH CrRW FW pOW NW pPMW dPMW CTS-R CTS-L CBL CZL ZW	Figure 1A Figure 1A Figure 1C Figure 1A,C Figure 1A Figure 1A Figure 1A Figure 1A Figure 1A Figure 1B Figure 1B Figure 1B Figure 1B Figure 1B
**1** **2** **3** **4** **5** **6** **7** **8** **9** **10** **11**	*Scapula* Dorsal border length (right or left) Cranial border length (right or left) Caudal border length (right or left) Maximum length of dorsal part (right or left) Maximum length of ventral part (right or left) Length of supraglenoid (right or left) Length of glenoid cavity (right or left) Width of scapula neck (right or left) Width of scapula head (right or left) Length of scapula spine (right or left) Length of suprahematus process (right or left)	DBL-R or DBL-L CrBL-R or CrBL-L CaBL-R or CaBL-L MLD-R or MLD-L MLV-R or MLV-L LS-R or LS-L LGC-R or LGC-L WSN-R or WSN-L WSH-R or WSH-L LSS-R or LSS-L LSP-R or LSP-L	Figure 1D Figure 1D Figure 1D Figure 1D Figure 1D Figure 1D Figure 1D Figure 1D Figure 1D Figure 1D Figure 1D
**1** **2** **3** **4** **5** **6** **7** **8** **9** **10** **11** **12** **13** **14** **15** **16**	*Pelvis* Horizontal width of pelvic girdle Horizontal width of pelvic aperture Horizontal width between the outer margins of the acetabula Width between eminentiae iliopubicae Length of pelvic girdle Length of the symphysis Diagonal height of pelvic aperture Width of ilium wing Length of ilium Minimum width of ilium shaft Minimum perimeter of ilium shaft Minimum perimeter of the pubis shaft Minimum perimeter of the ischium shaft Maximum inner length of foramen obturatum Maximum length of acetabulum Horizontal distance between outer margin of acetabulum and tuber coxae	HWPG HWPA HWA WEI LPG LPS DHPA WIW LIl WIS PIl PPS PIs LFO LA HAT	Figure 1F Figure 1F Figure 1F Figure 1D Figure 1F Figure 1D Figure 1D Figure 1D Figure 1F Figure 1F Figure 1F Figure 1D Figure 1D Figure 1D Figure 1D Figure 1D

**Table 2 biology-14-00933-t002:** Logistic regression models evaluating the predictive accuracy of individual pelvic morphometric ratios (LPG/LPS, LPG/PPS, and LPS/PPS) for sex determination.

	Coefficient B	Standard Error	z	*p*	Odds Ratio	95% Conf. Interval
Constant	−12.54	13.31	0.94	0.346	0	0–769,660.75
LPG/LPS	5.27	5.21	1.01	0.312	194.65	0.01–5,346,092.68
Constant	−7.6	6.64	1.14	0.253	0	0–225.81
LPG/PPS	1.7	1.32	1.28	0.199	5.48	0.41–73.54
Constant	−3.33	5.29	0.63	0.53	0.04	0–1149.56
LPS/PPS	2.19	2.71	0.81	0.419	8.95	0.04–1815.33

NOTE: None of the tested parameters reached statistical significance (*p* > 0.05). LPG/LPS exhibited the highest odds ratio (194.65), indicating a potential association with sex differentiation. However, the wide 95% confidence interval reflects considerable uncertainty. In parallel, LPG/PPS and LPS/PPS showed elevated odds ratios (5.48 and 8.95, respectively), though with non-significant *p*-values (0.199 and 0.419), and large confidence intervals.

**Table 3 biology-14-00933-t003:** The accuracy of sex prediction using LPG/LPS, LPG/PPS, or LPS/PPS.

		Female	Male	Correct
Observed	Female	8	0	100%
	Male	2	1	33.33%
	Total			81.82%

**Table 4 biology-14-00933-t004:** Logistic regression results for sex prediction in elephants using selected pelvic bone morphometric ratios.

	Coefficient B	Standard Error	z	*p*	Odds Ratio	95% Conf. Interval
Constant	−21.06	16.04	1.31	0.189	0	0–32,162.71
LPG/LPS	5.39	5.64	0.95	0.34	218.46	0–13,936,770.15
LPG/PPS	1.64	1.31	1.26	0.209	5.18	0.4–67.25
Constant	−7.48	6.61	1.13	0.258	0	0–239.81
LPG/PPS	4.58	3.13	1.46	0.144	97.12	0.21–44,861.43
LPS/PPS	−7.37	6.89	1.07	0.285	0	0–466.26

NOTE: Two separate model sets are presented: the first model includes the LPG/LPS and LPG/PPS, and the second model evaluates the LPG/PPS and LPS/PPS. Although none of the predictors has statistical significance (*p* > 0.05), the LPG/LPS in the first model yields a high odds ratio (218.46), indicating a strong association with sex classification. In the second model, the LPG/PPS also shows a relatively high odds ratio (97.12), with a *p*-value approaching significance (*p* = 0.144), suggesting potential predictive relevance. The wide confidence intervals in all predictors suggest variability in the estimates due to sample size limitations.

**Table 5 biology-14-00933-t005:** The accuracy of sex prediction using “LPG/LPS and LPG/PPS” or “LPG/PPS and LPS/PPS”.

		Female	Male	Correct
Observed	Female	7	1	87.5%
	Male	1	2	66.67%
	Total			81.82%

**Table 6 biology-14-00933-t006:** Logistic regression analysis of pelvic bone morphometric ratios for sex prediction in elephants.

	Coefficient B	Standard Error	z	*p*	Odds Ratio	95% Conf. Interval
Constant	−309.65	229.07	1.35	0.176	0	0–3.251 × 10^60^
LPG/LPS	126.47	91.87	1.38	0.169	8.389 × 10^54^	0–1.331 × 10^133^
LPG/PPS	−66.69	47.34	1.41	0.159	0	0–214,221,969,531.24
LPS/PS	162.42	117.5	1.38	0.167	3.457 × 10^70^	0–3.584 × 10^170^

NOTE: The table presents three selected morphometric ratios: LPG/LPS (length of pelvic girdle to length of pubic symphysis), LPG/PPS (length of pelvic girdle to prepubic symphysis), and LPS/PS (length of pubic symphysis to pelvic span). None of the predictors reach statistical significance (*p* > 0.05), although the LPS/PS showed the highest coefficient (B = 162.42) and a large odds ratio (3.457 × 10^70^). Wide confidence intervals for all variables indicate variability and uncertainty in effect estimates, due to a limited sample size.

**Table 7 biology-14-00933-t007:** The accuracy of sex prediction using a combination of LPG/LPS, LPG/PPS, and LPS/PPS.

		Female	Male	Correct
Observed	Female	8	0	100%
	Male	1	2	66.67%
	Total			90.91%

## Data Availability

Data is contained within this article.

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
