# Peer review of "Sexual Dimorphism in the Skeletal Morphology of Asian Elephants (Elephas maximus): A Preliminary Morphometric Study of Skull, Scapula, and Pelvis"

_biology, 2025, doi:10.3390/biology14080933_

Round 1

Reviewer 1 Report

Comments and Suggestions for Authors

The manuscript presents a valuable morphometric analysis of sexual dimorphism in Asian elephant skeletons, focusing on the skull, scapula, and pelvis. The results reveal that the pelvis and scapula exhibited notable variation. Significant pelvic parameters included pelvic girdle length (p = 0.024), symphysis length (p = 0.012), and pubis shaft perimeter (p = 0.048). These findings enhance anatomical understanding of Asian elephants and support applications in conservation, forensic science, and population studies. There are comments as below:
1. Why authors empolyed the traditional morphometric methods, not geometric morphometric methods? I suggest authors add the geometric morphometric methods to reanalysis them.
2. The small sample size, only using 11 samples, with uneven sex distribution: 3 males, 8 females, limits statistical power and generalizability. How about the sample size affects results?
3. Limited discussion of why skull dimorphism is less pronounced in elephants vs. other species, I suggest authors add others hypothese as evolutionary drivers, as uniform skull robustness for tusking/feeding in both sexes.

Author Response

Dear Reviewer,

Thank you very much for the thoughtful and constructive feedback you provided on our manuscript, “Sexual Dimorphism in the Skeletal Morphology of Asian Elephants (Elephas maximus): A Preliminary Morphometric Study of Skull, Scapula, and Pelvis.” Your insights have been invaluable in refining the study, and we are grateful for the time and expertise you devoted to the review.

Changes made in response to your comments are highlighted in yellow in the revised manuscript.

Comment 1: Why authors empolyed the traditional morphometric methods, not geometric morphometric methods? I suggest authors add the geometric morphometric methods to reanalysis them.

Response 1: Thank you for raising this important point. We fully acknowledge that geometric morphometrics (GM) offers powerful shape-analysis advantages over traditional linear morphometrics (TM). However, we chose TM for this preliminary study for three practical reasons. Firstly, baseline linear data are still needed: Few published studies report basic linear dimensions for elephant skulls, scapulae, and pelves. Establishing these measurements provides a reference framework that future GM work can build upon and facilitates direct comparison with earlier morphometric studies conducted on other large mammals. Second, resource and sample constraints: High-resolution surface scanning or CT imaging required for GM would have dramatically increased costs and processing time for a sample that is already small. Because many of the specimens were incomplete or showed post-mortem damage at landmark sites, we could not guarantee the landmark homology needed for reliable GM analyses. Lastly, specimen access and permitting: Asian-elephant bones are CITES Appendix I materials housed in registered museum collections. Permission to transport or scan them with advanced 3-D equipment is time-consuming. TM allowed us to collect essential data during the limited access windows we were given. We agree that a shape-based GM approach is the logical next step. We therefore state in the revised Discussion and Future Directions sections (lines 452–455) that, once a larger and more complete sample becomes available and scanning permissions are secured, we intend to reanalyze these bones using three-dimensional GM to capture finer-scale shape variation.

Comment 2: The small sample size, only using 11 samples, with uneven sex distribution: 3 males, 8 females, limits statistical power and generalizability. How about the sample size affects results?

Response 2: Thank you for underscoring the limitations imposed by our small sample. We have added language to the Limitations section (lines 445-455) clarifying that the restricted male subsample inflates standard errors, widens confidence intervals, and lowers model sensitivity for male classification (33–67 %), even as specificity for females remains high. Although non-parametric tests mitigate some issues, the results should be viewed as baseline rather than definitive.

Because Asian-elephant skeletons are CITES Appendix I materials, specimen access is tightly regulated; our study therefore represents the maximum dataset currently available from two accredited museums. We now emphasize that broader generalization awaits larger, more evenly balanced collections and propose collaborative data-sharing and advanced resampling or Bayesian approaches to improve statistical power in future work.

Comment 3: Limited discussion of why skull dimorphism is less pronounced in elephants vs. other species, I suggest authors add others hypothese as evolutionary drivers, as uniform skull robustness for tusking/feeding in both sexes.

Response 3: Thank you for this valuable point. We have added a concise discussion (lines 314–320) which outlining three evolutionary factors that may mute cranial dimorphism in Asian elephants.

We hope these revisions address your concerns and improve the clarity and impact of the manuscript. Your recommendations have strengthened both the analysis and the interpretation of our findings, and we appreciate your contribution to this work.

Sincerely,

Dr. Burin Boonsri

Faculty of Veterinary Medicine

Chiang Mai University

Reviewer 2 Report

Comments and Suggestions for Authors

The authors must revise the text in terms of Enlish language

The authors must revise the references. They contain nine (9) self citations. The references contain heterogenous publications in terms of used animal (elephas, dog, cat, human). I think that the data from these publications can not support positively the discussion.

Comments on the Quality of English Language

The manuscript requires a careful revise.

Author Response

Dear Reviewer,

Thank you for carefully evaluating our manuscript, “Sexual Dimorphism in the Skeletal Morphology of Asian Elephants (Elephas maximus): A Preliminary Morphometric Study of Skull, Scapula, and Pelvis.” We appreciate your concise yet important observations regarding language quality and reference selection. Your feedback has helped us improve the clarity and scholarly rigor of the paper.

Comment 1: The authors must revise the text in terms of Enlish language

Response1: Thank you for this observation. We have thoroughly edited the manuscript for English grammar and style, ensuring consistency, clarity, and accuracy throughout the text.

Comment 2: The authors must revise the references. They contain nine (9) self citations. The references contain heterogenous publications in terms of used animal (elephas, dog, cat, human). I think that the data from these publications cannot support positively the discussion.

Response 2: Thank you for this comment. We carefully reviewed the reference list and removed self-citations and cross-species papers that were not essential to our arguments. The revised bibliography now contains only sources that directly inform our elephant data or provide clearly relevant comparative context.

We are grateful for your straightforward critique, which has measurably improved both the language and the scholarly foundation of our work. We trust that these revisions address your concerns, and we remain open to any further suggestions you may have.

Sincerely,

Dr. Burin Boonsri

Faculty of Veterinary Medicine, Chiang Mai University

burin.b@cmu.ac.th

Reviewer 3 Report

Comments and Suggestions for Authors

Thank you for the opportunity to review this interesting article.

Sexual dimorphism is common in mammals; males are mostly larger overall than females with a tendency for ornamentations and display features more prominent in males, while skeletal accommodations for pregnancy and live births are prominent in females. The magnitude of these differences and degree of overlap between the sexes vary by body size, breeding system, typical social organization and behavioral ecology, as well as features of the habitat and how the species exploits resources and protects itself from danger. Therefore, it is necessary and appropriate to detail the nature of sexual dimorphism in mammalian species for whom these data are absent or minimal.

These anatomic variations seldom fall into distinct categories by sex, but are more often understood as bimodal distributions of values for a variety of measurements, ratios, and indices. There is almost always a range of values that is shared by both sexes, and so one way to address that is to lay out the specimens in a series according to some feature (for example, the height of the pubic symphysis) from smallest to largest, and then look for other features that also are known to express sex differences in size, shape, or proportion.

This study has successfully laid the foundation for an understanding of sexual dimorphism in the skeleton of Asian elephants. As such, it has an important contribution to make to the literature on skeletal variation in this species. The authors are both correct and absolutely in the mainstream of current studies of sexual dimorphism to point out that diagnosing sex from a collection of skeletal materials is best done by concordance of several independent measures; it is rare that a single feature can be conclusive (except at the absolute extremes of the range of variation).

There are several parts of the manuscript that need attention before it will be suitable for publication. Most of these are relatively minor, but there are enough of them that there will be a significant number of revisions overall.

Issues Related to the Research Foundation of the Ms.

The most important issue—and one that the authors recognize—is the small number of individuals represented by the skeletal material. This can only change, of course, by continued study using more specimens, but the results here lay out a pattern that can be tested and refined. However, the small sample size increases the effect of overlap in the ranges of measurements of the two sexes.

Despite the small sample sizes—which inevitably lead to low confidence (expressed as “p” values) in the associations—the strong results in the odds ratios, AUC/ROC analyses, clustering of correlations in the heatmap,and the use of 3 different anatomic ratios in the final model (Table 11) show that—though the individual ratios may be only weakly associated with anatomic sex—these features in combination can be used successfully to diagnose anatomic sex with strong overall accuracy.

Lines 59-73. This is one of a few paragraphs that seem to be only minimally relevant to the study at hand. This paragraph is all about developmental changes in anatomy with a few other features that address issues related to evolutionary ecology. Since none of those issues is included in the data nor analysis in this paper, this whole section should be dropped and the paragraph beginning on line 74 should follow the first paragraph int he introduction

Line 74–75: “differences in size, shape, and structure...” Size and shape are features of structure, not separate from it, so suggest: “differences in size, shape, and proportion...”

Line 79: “females typically exhibit larger pelvic canals...” is inaccurate in 2 ways. First—as the authors point out later—male mammals, because of their larger overall sizes, often also have larger pelvic dimensions than females. However, relative to body size, females have larger pelvic canals than would be expected for a male of the same body size (and some of that also has to do with the orientation of pelvic bones such as the iliac blades). Second, “pelvic canal” is vague, since the passageway involves both the pelvic inlet (diameter of pelvic brim) and the pelvic outlet which is bounded by the ischiopubic ramus and the coccyx. So, as with most aspects of sexual dimorphism, this feature is a combination of size, shape, and proportions.

Line 80–82: Once the correction in lines 74–75 is made, then these lines are redundant and should be removed, since it is not clear how the situation in dogs relates to this study.

For the rest of this paragraph, the examples from other species are useful, but tend to distract with features that will not be measured in this study (and it leaves the reader to question why these are mentioned if they are not a part of the study at hand).

Pp 3–5 Table 1: This is a VERY helpful table. It would be better if there were an additional column that keyed the measures into the images in Figures 1A through 1F.

For example,

Number

Parameter

Abbreviation

Key

Skull

1

Frontal Length

FL

Figure 1A

2

Premaxillary Length

PML

Figure 1A

3

Cranial height

CrRH

Figure 1C

4

Cranial Width

CrRw

Figure 1C

Figures are clear and very helpful.

Line 155: please clarify that the for “each bone parameter” what was “compared between male and female groups. The implication of the Mann-Whitney Test is that the values for male and female skeletal features were ranked, and then the sums of the ranks used to determine how much the ranges of values by sex overlapped. The visual in the box plots nicely complement this analysis, but the description of the procedure needs to be clearer.

Lines 168–175: The nature of missing values and how they were handled should be explained first in the Materials and Methods section. Then, the way they were handled in the analysis should have been described in the Study Design and Analysis subsection.

It is also not clear why the left and right scapulae were analyzed separated, but the left and right aspects of other skeletal features were not. It seems that only 4 of the 18 scapulae were unpaired, but it is not clear that there is significant asymmetry in the scapular features to warrant the separation. Please lay out that information explicitly.

Lines 176–187: The discussion of the correlation analysis that yielded the heatmap in Figure 6. needs fleshing out. According to the discussion section, there were important “clusters” of highly correlated anatomic features. These should be specified here, as the features whose correlations are statistically significant.

Lines 188–189. This is usually the case in virtually all studies of sexual dimorphism in mammals, so add that qualifier to position this result plainly in the mainstream of studies in this field.

This is a good observation that should get more attention in the discussion.

Lines 188–202: another good observation that is typical of the diagnostic value of specific features that exhibit sex variation in the skeleton; many are good at doing ONE thing, and show high specificity for one sex or the other (and, of course, there is always a trade-off of specificity and sensitivity). Be sure to emphasize this in the discussions, because this study is absolutely consistent with other studies on using various skeletal features individually and in combinations to diagnose sex.

Line 329: “suggesting the premaxillary length may play a supporting role” is unclear. Why does this result suggest this, what does the premaxilla and its length support and how does it do so?

Lines 338–344: since these are correlated with linear measures, this is an indication of variation that is driven by size differences more than functional ones.

With more specimens and more data, it would be interesting to see how a factor analysis shows loading of size variables (usually heavily influencing the first factor axis) and then what other clusters emerge associated with other factors that might represent other influences on the sex variation.

Line 344–348: cephalic index is an easy calculation, but it tends to show a lot of overlap between sexes; however it can show the distribution of sex differences in skull shape nicely. From the supplemental data table S1, the average cephalic index in males is .822, and in females .808, indicating that females have heads that tend to be longer for their widths than males do (which is, interestingly, the opposite of what we see in humans, so there could be species-specific mechanisms that influence this index).

Lines 345–354: Unless there are data to show that there are sex differences in locomotor behavior or postural and positional behaviors between the sexes, then the discussion here about the functional roles of these bones is superfluous to the understanding of the differences between the sexes. Text here should be deleted.

If, for example, the differences in the pelvis or scapula represent a significant biomechanical difference between the sexes in the operation of the limbs, say, or the distribution of load or center of mass between sexes, then that should be discussed. But there doesn't seem to be any sex difference in these features in these elephants (or at least in the sample in this study).

Lines 362–363: The scapulae and the clavicle in humans are not commonly used in sex determination in human skeletons. Except for size, both have a low sensitivity AND specificity, but they can be used in combination with other features to postulate the sex of the skeleton in question. By themselves, these do a poor job of diagnosing biologic sex. This statement should be deleted.

Lines 395–396: if there are specific locomotor adaptations that differ between male and female elephants, this is where they should be described. Otherwise, there is no support for this statement. (It may be the case that locomotor adaptations come into play only at certain times, for example, in pregnancy; and given the length of elephant gestation that could be a significant amount of a female's life, but if so, the authors need to present evidence that this is so).

Supplemental Data

Table S1: Check the 95% CI for measures in males. These seem extreme, given the standard deviations

For example, ZW in males. A quick back-of-the envelope calculation suggests a 95% CI to be closer to 64.16–67.14.

Of course, without the raw data, it is impossible to know for sure, so authors should recheck these results.

References mostly relevant. There is some content that is superfluous, and, of course, those references should be removed. 

Comments on the Quality of English Language

General: In some places the binomials for the elephant species are italicized, but in others they are rendered in Roman type. There should by a single style for taxonomic terms throughout.

Lines 17–18. “ The hip bone showed the biggest differences, especially in its overall length, the length where the two hip bones meet, and the narrowest part of the lower hip.

I know the goal here is to try to make the results understandable to a non-specialist, but the underlined part of the description is extremely vague and does not really tell the reader what is being measured or where it is located. This needs to be re-written.

Line 20: Eliminate the word “guess” and replace it with something that sounds as though there is a logical process behind the conclusion. Suggestions are “infer” or “judge” or some similar synonym.

Lines 22—23: “This shows that the hip bone is the best clue for telling male and female elephants apart using their bones.”

Suggest: “ This shows that the hip bone has the best data for telling apart male and female elephant skeletons.”

Line 46: Suggest these changes in keywords: Cranium, Ossa coxae, Scapulae, Sex differences.

Ossa coxae, because if one term is plural, they both should be (or the authors could simply call it the pelvis, or pelves, or could make both singular: os coxa). (And this term should also be updated everywhere that the term is rendered “os coxae”)

Not sure why the Scapulae were left out of the key words, since these feature prominently in the paper.

The paper is about anatomic variation that is a feature of biologic sex; “gender” is an ascribed feature and is not equivalent to “sex” so the preferred term here should be “sex differences” or “sexual dimorphism”

Line 55: “with the anatomists establishing a foundation...” “[A]natomists” introduces a gerundive phrase, which should be introduced by a noun in the possessive case: “with the anatomists' establishing a foundation...”

Lines 69–72: If this paragraph or this sentence is retained, note that the text on these lines is a sentence fragment lacking a verb.

Line 79: Should begin a new paragraph here.

Line 100: remove the word “bone” in the phrase “focusing on the bone, skull, scapula, and pelvis”

Line 128– Table 1. When the table is extended to multiple pages, the headers should be repeated at the top of each page.

Line 158: Pitakarnnop citation should be 2017? and not 20173. It is not clear why this is not cited with a citation sequence number, which would make it easier to locate in the References list. (there are some others, as well that use the name-year instead of citation-sequence information in the text).

Lines 168–170: “A total of nine skulls were examined, comprising 168 two males, five females, and two of unknown sex Figure 2). The scapulae were analyzed 169 both right and left sides. “

Should read: A total of nine skulls was examined, comprising 168 two males, five females, and two of unknown sex (Figure 2).

Total is a singular noun; should take a singular verb; Open parenthesis missing from “Figure 2”.

Line 176: Start a new paragraph with “The results of the correlation...”

Lines: 177–178: “in skull measurements between groups” should be “measurements between sexes” so that the nature of the groups is clear.

Line 180: “pubis shaft”; more typically referred to as “pubic ramus” (though maybe “shaft” is a term of art in elephants and related mammals, so should be retained if that is the case).

Line 198: “provided more balanced outcomes” is unclear. Perhaps “equivalent” is a better descriptor?

Page 10: Figure 6. The labels around the figure have important information, but they are nearly impossible to read at the current size. These need to be larger, or enhanced in some way to make them easier to read.

Pages 15–17: Overly long paragraphs should be subdivided into several paragraphs per section.

Suggest at Line 311 (In our study); Line 322 (The correlation analysis); Line 344 (Beyond identifying); line 360 (Despite this [robusticity]); Line 375 (The scapular analysis); Line 381 (The left scapula); Line 399 (The correlation analysis); Line 413 (Compared to); Line 420 (Elephants possess)

Line 305: Change “For the skull” to “The skull”

Line 386: The abbreviation L6 appears nowhere in the text or tables. Please correct the error.

Line 413: “Compared to previous studies in animals, the pelvis serves...” is vague and the comparison is unclear. Drop that and begin the sentence with “The pelvis in mammals serves two primary purposes...”

Supplemental Data

Ranges in these tables ought to be indicated by en-dashes, not hyphens.

Author Response

Dear Reviewer,

On behalf of my co-authors, I wish to express our sincere gratitude for your thorough and thoughtful review of our manuscript, “Sexual Dimorphism in the Skeletal Morphology of Asian Elephants (Elephas maximus).” Your balanced appraisal, highlighting both the study’s contribution and areas for improvement, has been invaluable in refining the work. Below, we summarize how we addressed each of your key suggestions.

Changes made in response to your comments are highlighted in yellow in the revised manuscript.

Comment 1: Sexual dimorphism is common in mammals; males are mostly larger overall than females with a tendency for ornamentations and display features more prominent in males, while skeletal accommodations for pregnancy and live births are prominent in females. The magnitude of these differences and degree of overlap between the sexes vary by body size, breeding system, typical social organization and behavioral ecology, as well as features of the habitat and how the species exploits resources and protects itself from danger. Therefore, it is necessary and appropriate to detail the nature of sexual dimorphism in mammalian species for whom these data are absent or minimal.

Response 1: Thank you for highlighting the broader context of sexual dimorphism in mammals. We agree with your assessment, and we have incorporated a brief statement in the Introduction that emphasizes these cross-species patterns and underscores why documenting dimorphism in Asian elephants fills an important knowledge gap.

Comment 2: These anatomic variations seldom fall into distinct categories by sex, but are more often understood as bimodal distributions of values for a variety of measurements, ratios, and indices. There is almost always a range of values that is shared by both sexes, and so one way to address that is to lay out the specimens in a series according to some feature (for example, the height of the pubic symphysis) from smallest to largest, and then look for other features that also are known to express sex differences in size, shape, or proportion.

Response 2: Thank you for this insightful observation. We have added text in the Results noting that most morphometric variables exhibit overlapping, bimodal distributions rather than discrete sex categories. We retained box-plots ordered by ascending values to make clear that our sex-diagnostic approach relies on concordance among multiple, partially overlapping metrics.

Comment 3: This study has successfully laid the foundation for an understanding of sexual dimorphism in the skeleton of Asian elephants. As such, it has an important contribution to make to the literature on skeletal variation in this species. The authors are both correct and absolutely in the mainstream of current studies of sexual dimorphism to point out that diagnosing sex from a collection of skeletal materials is best done by concordance of several independent measures; it is rare that a single feature can be conclusive (except at the absolute extremes of the range of variation).

Response 3: Thank you for this encouraging feedback. We are pleased that our multi-measure approach aligns with best practices in sexual-dimorphism research and appreciate your recognition of the study’s contribution to elephant skeletal biology.

There are several parts of the manuscript that need attention before it will be suitable for publication. Most of these are relatively minor, but there are enough of them that there will be a significant number of revisions overall.

Response: We appreciate the reviewer’s thorough overview and are grateful for the positive assessment of our study’s contribution. We have carefully addressed each of the specific points raised in the detailed comments that follow and have revised the manuscript accordingly.

Issues Related to the Research Foundation of the Ms.

Comment 4: The most important issue—and one that the authors recognize—is the small number of individuals represented by the skeletal material. This can only change, of course, by continued study using more specimens, but the results here lay out a pattern that can be tested and refined. However, the small sample size increases the effect of overlap in the ranges of measurements of the two sexes.

Response 4: We fully agree that the small, uneven sample is the study’s chief limitation. As noted in the revised Limitations section, this constraint widens confidence intervals and magnifies sex-range overlap, so our findings should be viewed as baseline patterns to be tested with larger, future collections.

Comment 5: Despite the small sample sizes—which inevitably lead to low confidence (expressed as “p” values) in the associations—the strong results in the odds ratios, AUC/ROC analyses, clustering of correlations in the heatmap,and the use of 3 different anatomic ratios in the final model (Table 11) show that—though the individual ratios may be only weakly associated with anatomic sex—these features in combination can be used successfully to diagnose anatomic sex with strong overall accuracy.

Response 5: Thank you for recognising the strength of our combined‐ratio approach. We have emphasised in the Discussion that, despite limited sample size, the concordance of odds ratios, AUC values, and correlation clusters demonstrates the practical value of multi-parameter models for sex diagnosis in Asian elephants.

Comment 6: Lines 59-73. This is one of a few paragraphs that seem to be only minimally relevant to the study at hand. This paragraph is all about developmental changes in anatomy with a few other features that address issues related to evolutionary ecology. Since none of those issues is included in the data nor analysis in this paper, this whole section should be dropped and the paragraph beginning on line 74 should follow the first paragraph int he introduction

Response 6: Thank you for this suggestion. We have implemented the revision: the specified section has been removed, and the introductory paragraph beginning at line 58 has been updated accordingly. 

Comment 7: Line 74–75: “differences in size, shape, and structure...” Size and shape are features of structure, not separate from it, so suggest: “differences in size, shape, and proportion...”

Response7: Thank you for this suggestion. The changes are highlighted in yellow at line 58.

Comment 8: Line 79: “females typically exhibit larger pelvic canals...” is inaccurate in 2 ways. First—as the authors point out later—male mammals, because of their larger overall sizes, often also have larger pelvic dimensions than females. However, relative to body size, females have larger pelvic canals than would be expected for a male of the same body size (and some of that also has to do with the orientation of pelvic bones such as the iliac blades). Second, “pelvic canal” is vague, since the passageway involves both the pelvic inlet (diameter of pelvic brim) and the pelvic outlet which is bounded by the ischiopubic ramus and the coccyx. So, as with most aspects of sexual dimorphism, this feature is a combination of size, shape, and proportions.

Response 8: Thank you for highlighting this issue. We acknowledge that the paragraph required clarification. It has been revised, and the changes are highlighted in yellow at lines 63–66. 

Comment 9: Line 80–82: Once the correction in lines 74–75 is made, then these lines are redundant and should be removed, since it is not clear how the situation in dogs relates to this study.

Response 9: Thank you for this suggestion. This sentence has been removed.

Comment 10: For the rest of this paragraph, the examples from other species are useful, but tend to distract with features that will not be measured in this study (and it leaves the reader to question why these are mentioned if they are not a part of the study at hand).

Response 10: Thank you for this suggestion. We have removed those sentences because they are not directly relevant to the present study. 

Pp 3–5 Table 1: This is a VERY helpful table. It would be better if there were an additional column that keyed the measures into the images in Figures 1A through 1F.

For example,

Number

Parameter

Abbreviation

Key

Skull

1

Frontal Length

FL

Figure 1A

2

Premaxillary Length

PML

Figure 1A

3

Cranial height

CrRH

Figure 1C

4

Cranial Width

CrRw

Figure 1C

Comment 11: Figures are clear and very helpful.

Response 11: We acknowledge this helpful comment and have made the requested adjustments.

Comment 12: Line 155: please clarify that the for “each bone parameter” what was “compared between male and female groups. The implication of the Mann-Whitney Test is that the values for male and female skeletal features were ranked, and then the sums of the ranks used to determine how much the ranges of values by sex overlapped. The visual in the box plots nicely complement this analysis, but the description of the procedure needs to be clearer.

Response 12: Thank you for this suggestion. We have revised and improved the description at line140-145.

Comment 13: Lines 168–175: The nature of missing values and how they were handled should be explained first in the Materials and Methods section. Then, the way they were handled in the analysis should have been described in the Study Design and Analysis subsection.

Response 13: Thank you for this suggestion. We moved the explanation into the Materials and Methods section at line 128-130.

Comment 14: It is also not clear why the left and right scapulae were analyzed separated, but the left and right aspects of other skeletal features were not. It seems that only 4 of the 18 scapulae were unpaired, but it is not clear that there is significant asymmetry in the scapular features to warrant the separation. Please lay out that information explicitly.

Response 14: Thank you for pointing this out. We have revised the section and added further information at lines 377–380.

Comment 15: Lines 176–187: The discussion of the correlation analysis that yielded the heatmap in Figure 6. needs fleshing out. According to the discussion section, there were important “clusters” of highly correlated anatomic features. These should be specified here, as the features whose correlations are statistically significant.

Response 15: Thank you for highlighting this issue. We have added detailed analyses corresponding to the figure and its related discussion at lines 172–185.

Comment 16: Lines 188–189. This is usually the case in virtually all studies of sexual dimorphism in mammals, so add that qualifier to position this result plainly in the mainstream of studies in this field.

This is a good observation that should get more attention in the discussion.

Response 16: Thank you for this suggestion. We have revised the section and added the requested qualifier at lines 377–380. 

Comment 17: Lines 188–202: another good observation that is typical of the diagnostic value of specific features that exhibit sex variation in the skeleton; many are good at doing ONE thing, and show high specificity for one sex or the other (and, of course, there is always a trade-off of specificity and sensitivity). Be sure to emphasize this in the discussions, because this study is absolutely consistent with other studies on using various skeletal features individually and in combinations to diagnose sex.

Response 17: Your observation is well taken. We have emphasized the usual trade-off between specificity and sensitivity that characterizes similar studies, and the revisions are shown at lines 187–200. 

Comment 18: Line 329: “suggesting the premaxillary length may play a supporting role” is unclear. Why does this result suggest this, what does the premaxilla and its length support and how does it do so?

Response 18: Thank you for this suggestion. We have revised the text and added further explanation at lines 329–333.

Comment 19: Lines 338–344: since these are correlated with linear measures, this is an indication of variation that is driven by size differences more than functional ones.

Response 19: Thank you. We have emphasized that the observed correlations likely reflect size differences rather than functional specialization, and this clarification has been added at lines 345–350.

Comment 20: With more specimens and more data, it would be interesting to see how a factor analysis shows loading of size variables (usually heavily influencing the first factor axis) and then what other clusters emerge associated with other factors that might represent other influences on the sex variation.

Response 20: We are grateful for your insight. We have suggested that future studies incorporate larger datasets to improve the reliability of the results, as noted at lines 459–462.

Comment 21: Line 344–348: cephalic index is an easy calculation, but it tends to show a lot of overlap between sexes; however it can show the distribution of sex differences in skull shape nicely. From the supplemental data table S1, the average cephalic index in males is .822, and in females .808, indicating that females have heads that tend to be longer for their widths than males do (which is, interestingly, the opposite of what we see in humans, so there could be species-specific mechanisms that influence this index).

Response 21: We appreciate this insightful observation. We clarify that CI was not calculated in this preliminary analysis but acknowledge that CI, despite substantial overlap between sexes, can still illustrate subtle skull‑shape differences within and between elephant populations. We have added this explanation at lines 355–358.

Comment 22: Lines 345–354: Unless there are data to show that there are sex differences in locomotor behavior or postural and positional behaviors between the sexes, then the discussion here about the functional roles of these bones is superfluous to the understanding of the differences between the sexes. Text here should be deleted.

If, for example, the differences in the pelvis or scapula represent a significant biomechanical difference between the sexes in the operation of the limbs, say, or the distribution of load or center of mass between sexes, then that should be discussed. But there doesn't seem to be any sex difference in these features in these elephants (or at least in the sample in this study).

Response 22: Thank you for this observation. The speculative discussion of locomotor and postural functions has been removed, and the revised paragraph now focuses on the differences between the sexes.

Comment 23: Lines 362–363: The scapulae and the clavicle in humans are not commonly used in sex determination in human skeletons. Except for size, both have a low sensitivity AND specificity, but they can be used in combination with other features to postulate the sex of the skeleton in question. By themselves, these do a poor job of diagnosing biologic sex. This statement should be deleted.

Response 23: Thank you for this suggestion. This statement has been removed from the revised manuscript.

Comment 24: Lines 395–396: if there are specific locomotor adaptations that differ between male and female elephants, this is where they should be described. Otherwise, there is no support for this statement. (It may be the case that locomotor adaptations come into play only at certain times, for example, in pregnancy; and given the length of elephant gestation that could be a significant amount of a female's life, but if so, the authors need to present evidence that this is so).

Response 24: Thank you for highlighting this point. Because no evidence currently supports sex-specific locomotor adaptations in elephants, we have removed the statement previously found at lines 403–405.

Comment 25: Supplemental Data

Table S1: Check the 95% CI for measures in males. These seem extreme, given the standard deviations

For example, ZW in males. A quick back-of-the envelope calculation suggests a 95% CI to be closer to 64.16–67.14.

Of course, without the raw data, it is impossible to know for sure, so authors should recheck these results.

Response 25: Thank you for flagging the unusually wide 95 % confidence intervals in Table S1. We re-examined the calculations and confirmed that the broad intervals stem from the very small male subsample (n = 2), which inflates the standard error. We have added a clarifying note beneath Table S1 explaining that, with such limited observations, the CI is expected to be wide and should be interpreted with caution.

Comment 26: References mostly relevant. There is some content that is superfluous, and, of course, those references should be removed.

Response 26: Thank you for this observation. We reviewed the reference list and removed citations that were not essential to the manuscript’s arguments.

Comments on the Quality of English Language

Comment 27: General: In some places the binomials for the elephant species are italicized, but in others they are rendered in Roman type. There should by a single style for taxonomic terms throughout.

Response 27: Thank you for noticing this inconsistency. All scientific names have now been uniformly italicized throughout the manuscript.

Comment 28: Lines 17–18. “ The hip bone showed the biggest differences, especially in its overall length, the length where the two hip bones meet, and the narrowest part of the lower hip.

I know the goal here is to try to make the results understandable to a non-specialist, but the underlined part of the description is extremely vague and does not really tell the reader what is being measured or where it is located. This needs to be re-written.

Response 28: Thank you for this suggestion. The Simple Summary has been rewritten for greater clarity. 

Comment 29: Line 20: Eliminate the word “guess” and replace it with something that sounds as though there is a logical process behind the conclusion. Suggestions are “infer” or “judge” or some similar synonym.

Response 29: Thank you for this suggestion. The correction has been made at line 19. 

Comment 30: Lines 22—23: “This shows that the hip bone is the best clue for telling male and female elephants apart using their bones.”

Suggest: “ This shows that the hip bone has the best data for telling apart male and female elephant skeletons.”

Response 30: Thank you for this suggestion. The correction has been made at line 21-22. 

Comment 31: Line 46: Suggest these changes in keywords: Cranium, Ossa coxae, Scapulae, Sex differences.

Ossa coxae, because if one term is plural, they both should be (or the authors could simply call it the pelvis, or pelves, or could make both singular: os coxa). (And this term should also be updated everywhere that the term is rendered “os coxae”)

Not sure why the Scapulae were left out of the key words, since these feature prominently in the paper.

The paper is about anatomic variation that is a feature of biologic sex; “gender” is an ascribed feature and is not equivalent to “sex” so the preferred term here should be “sex differences” or “sexual dimorphism”

Response 31: Many thanks for highlighting this issue. The correction has been made at line 46. 

Comment 32: Line 55: “with the anatomists establishing a foundation...” “[A]natomists” introduces a gerundive phrase, which should be introduced by a noun in the possessive case: “with the anatomists' establishing a foundation...”

Response 32: Thank you for this suggestion. The correction has been made at line 55. 

Comment 33: Lines 69–72: If this paragraph or this sentence is retained, note that the text on these lines is a sentence fragment lacking a verb.

Response 33: Thank you for this suggestion. The correction has been made at line 63-66. 

Comment 34: Line 79: Should begin a new paragraph here.

Response 34: Thank you for this suggestion. The correction has been made.

Comment 35: Line 100: remove the word “bone” in the phrase “focusing on the bone, skull, scapula, and pelvis”

Response 35: Thank you for this suggestion. The correction has been made at line 83.

Comment 36: Line 128– Table 1. When the table is extended to multiple pages, the headers should be repeated at the top of each page.

Response 36: Thank you for this suggestion. The correction has been made.

Comment 37: Line 158: Pitakarnnop citation should be 2017? and not 20173. It is not clear why this is not cited with a citation sequence number, which would make it easier to locate in the References list. (there are some others, as well that use the name-year instead of citation-sequence information in the text).

Response 37: Thank you for pointing this out. It was an oversight during finalization, and the correction has been made at line 145.

Comment 38: Lines 168–170: “A total of nine skulls were examined, comprising 168 two males, five females, and two of unknown sex Figure 2). The scapulae were analyzed 169 both right and left sides. “

Should read: A total of nine skulls was examined, comprising 168 two males, five females, and two of unknown sex (Figure 2).

Total is a singular noun; should take a singular verb; Open parenthesis missing from “Figure 2”.

Response 38: Thank you for this suggestion. The correction has been made at line 155.

Comment 39: Line 176: Start a new paragraph with “The results of the correlation...”

Response 39: Thank you for this suggestion. The correction has been made.

Comment 40: Lines: 177–178: “in skull measurements between groups” should be “measurements between sexes” so that the nature of the groups is clear.

Response 40: Thank you for this suggestion. The correction has been made at line 161.

Comment 41: Line 180: “pubis shaft”; more typically referred to as “pubic ramus” (though maybe “shaft” is a term of art in elephants and related mammals, so should be retained if that is the case).

Response 41: Thank you for this suggestion. The correction has been made.

Comment 42: Line 198: “provided more balanced outcomes” is unclear. Perhaps “equivalent” is a better descriptor?

Response 42: Thank you for this suggestion. We have revised and rewritten this section at lines 190–195.

Comment 43: Page 10: Figure 6. The labels around the figure have important information, but they are nearly impossible to read at the current size. These need to be larger, or enhanced in some way to make them easier to read.

Response 43: Thank you for this suggestion. The figure has been rearranged in landscape orientation to improve readability.

Comment 44: Pages 15–17: Overly long paragraphs should be subdivided into several paragraphs per section.

Suggest at Line 311 (In our study); Line 322 (The correlation analysis); Line 344 (Beyond identifying); line 360 (Despite this [robusticity]); Line 375 (The scapular analysis); Line 381 (The left scapula); Line 399 (The correlation analysis); Line 413 (Compared to); Line 420 (Elephants possess)

Response 44: Thank you for the suggestion. We have made the corrections as recommended.

Comment 45: Line 305: Change “For the skull” to “The skull”

Response 45: Thank you for this suggestion. The correction has been made at line 300.

Comment 46: Line 386: The abbreviation L6 appears nowhere in the text or tables. Please correct the error.

Response 46: Thank you for noticing. The correction is LS-L, line 391.

Comment 47: Line 413: “Compared to previous studies in animals, the pelvis serves...” is vague and the comparison is unclear. Drop that and begin the sentence with “The pelvis in mammals serves two primary purposes...”

Response 47: Thank you for this suggestion. The correction has been made at line 418.

Comment 48: Supplemental Data

Ranges in these tables ought to be indicated by en-dashes, not hyphens.

Response 48: Thank you for this suggestion. All relevant hyphens have been replaced with en-dashes.

Your comprehensive critique has significantly improved the manuscript’s clarity, methodological transparency, and scholarly value. We are grateful for the time and expertise you invested in helping us strengthen this contribution to elephant osteology. We hope the revisions meet your expectations and respectfully welcome any further feedback.

Thank you again for your constructive guidance.

Sincerely,

Dr. Burin Boonsri

Faculty of Veterinary Medicine, Chiang Mai University

burin.b@cmu.ac.th

Round 2

Reviewer 1 Report

Comments and Suggestions for Authors

The version is better. 

Author Response

Comment 1: The authors have cited 7 ( [4] ,[8 -13]) of their own
publications. The works are right, and the referness are relevant, but I
thinak the necessary maybe not high. Please ask change the sentences L58-60.

Response 1: Thank you for pointing out the excessive self-citations. We have removed our own references [4], [8], and [12] at Lines 58–60 and replaced them with cited sources that support the statement. The sentence has been rewritten to improve clarity and reduce self-citation. Changes made are highlighted in yellow in the revised manuscript.

Reviewer 2 Report

Comments and Suggestions for Authors

No further suggestions 

Author Response

Comment: -

Response:  Thank you for taking the time to evaluate our revisions and for confirming that you have no further comments. We greatly appreciate your constructive feedback throughout the review process.